# Cancer in HIV-positive and HIV-negative adolescents and young adults in South Africa: a cross-sectional study

Tafadzwa Dhokotera ,[1,2,3,4,5] Julia Bohlius,[2,3,4] Matthias Egger ,[2,6,7] Adrian Spoerri,[2] Jabulani Ronnie Ncayiyana ,[5,6] Gita Naidu,[8] Victor Olago ,[1,5] Marcel Zwahlen ,[2] Elvira Singh,[1,5] Mazvita Muchengeti [1,5,9]

For numbered affiliations see end of article.

**Correspondence to**
Tafadzwa Dhokotera;
tafadzwagladys.dhokotera@swisstph.ch

## ABSTRACT

**Objective** To determine the spectrum of cancers in adolescents and young adults (AYAs) living with and without HIV in South Africa.

**Design** Cross-sectional study with cancer records provided by the National Cancer Registry (NCR) and HIV records from the National Health Laboratory Service (NHLS).

**Setting and participants** The NHLS is the largest provider of pathology services in the South African public sector. The NCR is a division of the NHLS. We included AYAs (aged 10–24 years) diagnosed with cancer by public health sector laboratories between 2004 and 2014 (n=8479). HIV status was obtained through record linkages and text mining.

**Primary and secondary outcomes** We determined the spectrum of cancers by HIV status in AYAs. We used multivariable logistic regression to describe the association of cancer in AYAs with HIV, adjusting for age, sex, ethnicity and calendar period. We imputed (post hoc) the HIV status for AYA with unknown HIV status.

**Results** 8479 AYAs were diagnosed with cancer, HIV status was known for 45% (n=3812). Of those whose status was known, about half were HIV positive (n=1853). AYAs living with HIV were more likely to have Kaposi's sarcoma (adjusted OR (aOR) 218, 95% CI 89.9 to 530), cervical cancer (aOR 2.18, 95% CI 1.23 to 3.89), non-Hodgkin's lymphoma (aOR 2.12, 95% CI 1.69 to 2.66) and anogenital cancers other than cervix (aOR 2.73, 95% CI 1.27 to 5.86) than AYAs without HIV. About 44% (n=1062) of AYAs with HIV-related cancers had not been tested for HIV.

**Conclusions** Targeted HIV testing for AYAs diagnosed with cancer, followed by immediate start of antiretroviral therapy, screening for cervical precancer and vaccination against human papilloma virus is needed to decrease cancer burden in AYAs living with HIV in South Africa.

## STRENGTHS AND LIMITATIONS OF THIS STUDY

⇒ This is the first nationwide study in South Africa to compare the distribution of cancers in adolescents and young adults (AYAs) by HIV status.
⇒ The record linkage and the additional results determined from the text mining process ensured that we extracted the maximum available HIV results.
⇒ The record linkage and the additional results determined from the text mining process ensured that we extracted the maximum available HIV results.
⇒ We assumed a CD4 cell count test indicates being HIV positive but CD4 testing maybe performed for other reasons.
⇒ Since this was a population of only AYAs diagnosed with cancer, the ORs could be overestimated or underestimated depending on the frequency of the cancer.

the early days of the HIV epidemic.[3 4] The outcomes of AYAs living with HIV (AYALHIV) have been poor. Challenges to retain AYAs in care and lack of adherence may lead to poor virologic suppression and low CD4 cell counts, which endangers their health.[3 5–7] Coinfection with other oncogenic viruses is also common in this age group.[8 9] For people living with HIV, immunodeficiency and coinfections with other oncogenic viruses are risk factors for developing cancer.

Several studies have shown that the risk of HIV-related cancers—non-Hodgkin's lymphoma (NHL), Kaposi's sarcoma (KS) and cervical cancer (CC) is higher in AYALHIV than in HIV negative AYA.[10–15] In the USA, the incidence of leiomyosarcoma was also higher in AYALHIV than in their peers from the general population.[10] However, most of the existing data are from settings with a low HIV burden, but we still know little about cancer burden and risk in AYALHIV in high HIV burden African countries, like South Africa. Estimating the relationship between cancer and HIV is important to determine

## INTRODUCTION

In Eastern and Southern Africa, an estimated 1.2 million adolescents aged 10–19 years are living with HIV.[1] Young people aged 15–25 years represent 30% of new infections.[2] Children infected with HIV perinatally are now more likely to live and to become adolescents and young adults (AYA) compared with

their additional healthcare needs and to provide insights on potential mechanisms for prevention of cancer development in AYALHIV.

We aimed to evaluate the spectrum and cancers associated with HIV in AYAs at a national level. The South African HIV Cancer Match (SAM) study was created to identify the risk factors and spectrum of malignancies in people living with HIV based on routine reports.[16] In this cross-sectional analysis, which is a subproject of the SAM study, we included all AYAs with a pathology-confirmed cancer diagnosis. We examined the proportion of cancer diagnoses with or without HIV infection and the factors associated with cancer in AYALHIV.

## METHODS

### Study design and setting

This was a cross-sectional study with cancer data provided by the National Cancer Registry (NCR) and HIV-related laboratory data from the National Health Laboratory Service (NHLS). The NHLS is the largest provider of diagnostic pathology services in the South African public sector (estimated coverage is over 80% of the South African population).[17] The NHLS includes the National Institute of Communicable Diseases, the National Institute of Occupational Health and the pathology-based NCR. The Corporate Data Warehouse (CDW) is the centralised data centre of the NHLS where all the data on tests performed in its laboratories are stored.

### Inclusion criteria

We included all AYAs with a primary incident cancer recorded from 2004 to 2014 in NCR records irrespective of HIV status. Adolescence was defined as 10–19 years and young adulthood as 20–24 years at the time of cancer diagnosis, based on WHO and South African Department of Health definitions.[6 18] We excluded cancer precursors and only retained laboratory-confirmed cancer records that contained the International Classification of Disease in Oncology V.3 topography and morphology descriptions.[19] If a person had two different cancers at different sites, they were considered as two individual records (multiple primaries).

### Outcome and exposure variables

The main exposure was HIV infection and the main outcome cancer diagnosis stratified by morphological type and subtype where applicable. HIV status was determined from HIV diagnostic tests (enzyme-linked immunosorbent assay, qualitative PCR and rapid HIV tests) and HIV monitoring tests (CD4 cell counts and HIV RNA viral loads). We assumed an individual was HIV positive if any diagnostic test was positive or if monitoring tests (CD4 cell count, HIV RNA viral load) were recorded. We used text mining methods to extract additional HIV results from the clinical history section of cancer pathology reports as discussed in detail elsewhere.[20] We assigned the HIV status irrespective of the cancer diagnosis date.

We used deterministic and probabilistic record linkages (PRL) as well as text mining to determine HIV status. For the deterministic record linkage, we used episode numbers as linkage variable. Episode number refers to tests that were requested for a patient at the same time by the health practitioner and assigned the same unique identifier. About 65% of the all linkages were matched using the episode number. Using PRL, the CDW created a unique patient identifier for records belonging to the same person. As described in detail elsewhere.[20] The CDW uses names, surnames and dates of births as linkage variables, which are fed into the PRL linkage algorithm. First names and surnames have a weight of 40% each, and date of birth a weight of 20%. Records with a recorded national identity number are exact matches. To be considered a match, the cumulative score has to reach 90% or above. The data from the CDW have been evaluated for completeness and accuracy and validated as a good source of data for research on HIV in South Africa.[21] After this, we then added the text mining data. We used NCR records to determine demographic characteristics. Where missing, the NCR imputes ethnicity based on surnames using known surname-ethnicity pairings.[22] Ethnicity was grouped into Black and Other for comparison purposes because few subjects belonged to other population subgroups. We divided calendar years into three periods: the early years of combination antiretroviral therapy (ART) (2004–2008); later years (2009–2011); and, the most recent period (2012–2014). We selected cut-offs for the calendar periods to correspond with changes in ART guidelines in South Africa during the study period.[23] We grouped cancers of vulva, penis, vagina and anus as anogenital cancers other than CC. We evaluated NHL as a group, and at each of its subtypes: Burkitt lymphoma; diffuse large B cell lymphoma (DLBCL); diffuse immunoblastic large B cell lymphoma (DILBCL); follicular NHL and mature T cell NHL.

### Data analysis

For descriptive purposes, we presented sex, ethnicity (Asian, black, coloured (mixed race) and white) and age stratified by HIV status (positive/negative/unknown). We then showed the frequency and spectrum of the top 20 cancers in AYAs stratified by HIV status (positive/negative/unknown) and by sex. We used a logistic regression model to determine the association between HIV and cancer in AYAs. For each cancer, we used records without the cancer under study as the comparison group, including cancers with an infectious aetiology. We adjusted the models using age (adolescence vs young adults), sex (male vs female, except for sex-specific cancers), ethnicity (black vs other) and ART era. We restricted our main analyses to cancers in AYAs with known HIV status, so all AYA were either HIV positive or HIV negative. We assessed interactions between HIV and other factors of interest (age, sex, ethnicity and calendar period) and adjusted models for interaction analysis for age, sex, ethnicity and calendar period. To test for significance of

the interaction, we used likelihood ratio tests to compare logistic regression models with and without the interaction terms at 5% significance level. Stata V.15.1 was used for all analyses (StataCorp).

### Sensitivity analysis

As a sensitivity analysis, we used multiple imputation methods to impute missing HIV results for 4431 patients with cancer with unknown HIV status. We included HIV status (the primary exposure), cancer diagnosis period, sex, age and ethnicity in our imputation model. Since ethnicity was already imputed by the NCR using surname-ethnicity pairings, we excluded records that still had missing ethnicity data (4%; n=368). We also excluded records with missing sex as they were few (0.09%; n=8). We used multivariable imputation with chained equations to generate 15 imputed datasets that we combined to give a pooled estimate (OR). For each cancer, we fitted multivariable logistic regression models adjusting for age, sex, ethnicity and calendar period. We compared the results from the imputed dataset with the main complete case analysis. Online supplemental table S1 compares the proportional distribution of known and unknown HIV status by the variables in the imputation model.

### Patient and public involvement

The study is based on routinely collected laboratory data therefore no patients were involved in the design, conduct, reporting, or dissemination plans of our research. Due to the anonymous nature of the data, we cannot disseminate the results of analyses of the data directly to study participants.

### RESULTS

Over the 11 years of study period, 8479 AYAs were diagnosed with cancer. The median age was 20 years (IQR: 15–23) and over half (n=4466) of all recorded cancer cases were diagnosed in young adults (20–24 years). Girls and women were 54% (n=4 605) of the AYA population; most AYAs with cancer were black 75% (n=6376). About 45% (n=3819) of AYAs with cancer were assigned an HIV status; half of those with known status were HIV positive (n=1855). When we compared AYA patients with cancer with and without HIV, the median age of AYA patients with cancer with HIV was 22 years (IQR: 19–23) while it was 18 years in those without HIV (IQR: 13–21). In our analysis, AYAs with HIV were more often female (67% vs 45%; p<0.001) and more often black population (86% vs 64%) (table 1) as compared with AYA without HIV. The proportion of AYA with unknown HIV status declined across the calendar periods (figure 1).

The most frequently diagnosed cancer was KS, followed by leukaemia and bone cancer (figure 2, absolute numbers in online supplemental table S2). Non-AIDS defining cancers (NADCs) made up 68% (n=5803) of histologically diagnosed cancers. In AYA with AIDS defining cancers (ADCs, 44% (n=1062) of patients had

unknown HIV status vs 59% (n=3411) of AYA with NADC. The HIV status of 44% (n=617) of AYA diagnosed with KS was unknown, and the HIV status of 43% (n=269) of AYA diagnosed with NHL was unknown (figure 1). Haematological cancers were the most common cancers in AYAs without HIV: leukaemia was the most frequent diagnosis (n=449), followed by Hodgkin's lymphoma (n=246). Bone cancers were also more common in AYAs without HIV (n=197). In HIV negative AYAs, the top five cancers were similar for male and female patients but HIV negative male AYAs had a higher proportion of Hodgkin's lymphoma and bone cancers compared with female AYAs.

Among those with recorded HIV status, KS, NHL and Hodgkin's lymphoma, leukaemia and CC were the most frequent cancers in AYALHIV (figure 1). The top five most frequent cancers among female AYAs with HIV were KS, NHL, CC, Hodgkin's lymphoma and leukaemia (online supplemental figure S1). For male AYAs with HIV, the most frequently diagnosed cancers were KS, NHL, leukaemia, Hodgkin's lymphoma and connective tissue cancers. The proportion of KS cases was higher in female AYAs with HIV (71%, n=998) than in male AYAs with HIV (29%, n=409).

The logistic regression analysis revealed higher odds of ADCs than to NADCs in AYAs with HIV (table 2). When we compared HIV positive AYAs to HIV negative AYAs, the adjusted OR for AYAs with HIV was 218 (95% CI 89.9 to 530) for KS, 2.18 (95% CI 1.23 to 3.89) for CC, and 2.12 (95% CI 1.69 to 2.66) for NHL. The odds of specific NHL subtypes like Burkitt lymphoma, DLBCL and DILBCL were higher in AYALHIV than in AYAs without HIV (table 2). Anogenital cancers other than CC were also strongly associated with HIV; adjusted OR was 2.73 (95% CI 1.27 to 5.86). We did not observe significant odds of leiomyosarcoma in AYALHIV but, of the eight recorded leiomyosarcoma cases with a known HIV result, six were HIV positive and five were female. Odds were not high for HIV and Hodgkin's lymphoma or HIV and liver cancer.

Interaction tests determined that age modified the odds of NHL in AYALHIV; adolescents with HIV had higher odds of NHL (adjusted OR 3.17; 95% CI 2.35 to 4.28) than young adults with HIV (adjusted OR 1.29; 95% CI 0.93 to 1.79; p value for interaction <0.0001). Ethnicity also modified the odds of Burkitt lymphoma in HIV positive AYAs; black AYAs with HIV had higher odds of Burkitt lymphoma (adjusted OR 3.84; 95% CI 2.10 to 7.04) than non-black AYAs with HIV (adjusted OR 1.35; 95% CI 0.43 to 4.28, p value for interaction=0.0199). In the sensitivity analysis that used the imputed dataset multivariable analysis results were similar to the main analysis of subjects with a known HIV status (table 2). Specifically the odds of KS (adjusted OR 208; 95% CI 83.9 to 519, CC (adjusted OR 2.70; 95% CI 1.31 to 2.17) and anogenital cancers other than cervix (adjusted OR 2.61; 95% CI 1.11 to 6.11) were higher in AYALHIV compare to HIV negative AYAs.

**Table 1** Demographic characteristics of adolescents and young adults with a cancer diagnosis stratified by HIV status in the South African public health sector, 2004–2014

|  | HIV negative | HIV positive | HIV unknown |
|---|---|---|---|
|  | n (%) | n (%) | n (%) |
| Age |  |  |  |
| Median age (IQR) (years) | 18 (13–21) | 22 (19–23) | 20 (16–22) |
| 10–14 | 635 (32.3%) | 200 (10.8%) | 922 (19.8%) |
| 15–19 | 585 (29.8%) | 338 (18.2%) | 1331 (28.6%) |
| 20–24 | 744 (37.9%) | 1317 (71.0%) | 2407 (51.7%) |
| Sex |  |  |  |
| Female | 877 (44.7%) | 1247 (67.2%) | 2484 (53.3%) |
| Male | 1087 (55.3%) | 607 (32.7%) | 2169 (46.5%) |
| Missing | 0 (0%) | 1 (0.1%) | 7 (0.2%) |
| Ethnicity |  |  |  |
| Asian | 34 (1.7%) | 14 (0.8%) | 106 (2.3%) |
| Black | 1258 (64.1%) | 1593 (85.9%) | 3525 (75.6%) |
| Coloured (mixed race) | 323 (16.4%) | 103 (5.6%) | 317 (6.8%) |
| White | 274 (14.0%) | 74 (4.0%) | 487 (10.5%) |
| Missing | 75 (3.8%) | 71 (3.8%) | 225 (4.8%) |
| Type of cancer |  |  |  |
| Non-AIDS defining cancer | 1699 (86.5%) | 697 (37.6%) | 3411 (73.2%) |
| AIDS defining cancer | 206 (10.5%) | 1129 (60.9%) | 1062 (22.8%) |
| Primary site unknown | 59 (3.0%) | 29 (1.6%) | 187 (4.0%) |
| ART calendar period |  |  |  |
| 2004–2007 | 594 (30.2%) | 500 (27.0%) | 2062 (44.2%) |
| 2008–2011 | 822 (41.9%) | 784 (42.3%) | 1647 (35.3%) |
| 2012–2014 | 548 (27.9%) | 571 (30.8%) | 951 (20.4%) |
| Multiple primary cancer |  |  |  |
| Yes | 10 (0.50%) | 13 (0.70%) | 31 (0.67%) |
| No | 1954 (99.5%) | 1842 (99.3%) | 4629 (99.3%) |
| Total | 1964 (100%) | 1855 (100%) | 4660 (100%) |

Multiple primary cancer refers to an individual with more than one cancer at different primary sites.
ART, antiretroviral therapy; IQR, interquartile range
.

## DISCUSSION

We observed an association among AYAs between HIV and ADCs and anogenital cancers other than CC, including penile, anal, vulvar and vaginal cancers. Among those living with HIV, the proportion of KS was higher in girls and young women than in boys and young men. The combined odds of cancers not associated with HIV were higher in AYALHIV than in those without HIV. We could not ascertain the HIV status of many AYAs diagnosed with HIV-related cancers, however, a sensitivity analyses using imputed data yielded qualitatively similar results. We observed higher odds of Burkitt lymphoma black AYALHIV compared with those without HIV and higher odds of NHL in adolescents living with HIV compared with young adults living with HIV.

It is known that the risk of ADCs is higher in AYALHIV.[10–12 14 24 25] In our study, KS was the cancer most strongly associated with HIV. HIV cohort studies have reported increased KS incidence among children and adolescents under 16.[10 12] A multicohort study found KS risk was higher in HIV positive adolescents and children from Southern Africa than in the same age group in other regions of the world.[26] In South Africa, where treatment and retention in care rates for AYAs with HIV are low,[3] poorly controlled HIV infection among AYAs may increase the odds of KS. The South African National HIV Prevalence Survey of 2017 revealed that about 60% of young adults (ages 15–24) living with HIV were not on ART.[27] Untreated AYALHIV are likely to develop immunodeficiency which increases their risk of developing KS.[26] The higher proportion of KS in females in our study might also be a reflection of the high proportion of HIV observed in female AYAs in our study.

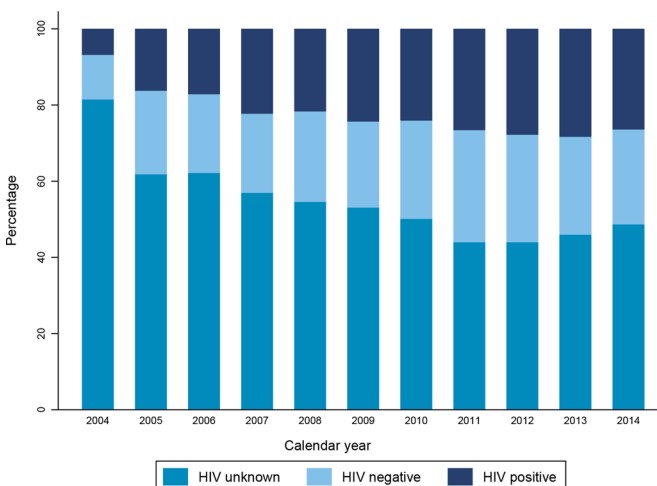

**Figure 1** Distribution of HIV unknowns across the study period among AYAs with cancer. The trend analysis for proportions was statistically significant across all strata of HIV status (p<0.001) for all HIV status stratified by year of cancer diagnosis. AYAs, adolescents and young adults.

The odds of CC and human papilloma virus (HPV) related cancers such as anogenital cancers in this young adult population may be increased for several reasons. In South Africa in 2017, girls and young women were much more likely to be HIV positive (10.9% prevalence) than boys and young men (4.8%).[27] Biological factors may account for higher HIV prevalence in girls and young women, along with socioeconomic factors that encourage risky sexual behaviour including transactional and intergenerational sexual relationships.[27] High prevalence and poorly controlled HIV can increase the risk of HPV coinfection in an age group less likely to be screened for precancerous cervical lesions, which in turn increases CC risk and risk of other anogenital cancers among AYAs. A study in the Western Cape province of South Africa found AYALHIV had higher HPV prevalence and more high-risk HPV subtypes than AYAs without HIV.[8] In contrast

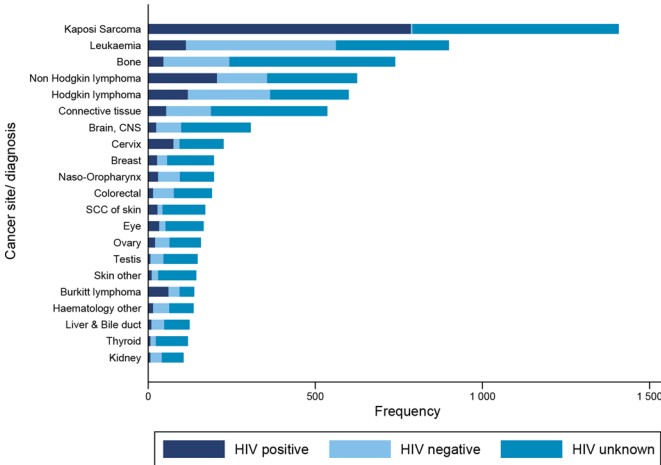

**Figure 2** Top 21 cancer in adolescents and young adults in the South African public health sector stratified by HIV status. Brain, CNS, brain central nervous system; SCC of skin, squamous cell carcinoma of skin.

to other studies on cancer in AYALHIV, we observed three CC cases in AYAs aged between 14 and 16 years; two of these young women were HIV positive. HIV cohorts in South Africa and the USA have not identified CC in children and adolescents under 16 years old,[10 14] but CC risk and incidence has been on the increase in the ART era for those between 18 and 24 years old.[28] Early sexual debut, and subsequent early exposure to causative agents like HPV may explain this early presentation with CC in South Africa.[8 29 30]

Lymphomas are often misdiagnosed as tuberculosis in people living with HIV in our setting, slowing lymphoma diagnosis and worsening patients' prognosis.[31] This might explain the significantly lower odds of Hodgkin lymphoma, a cancer associated with HIV in our study population. Like other studies, we found NHL was associated with HIV.[10 14] NHL is associated with poor adherence to ART and low rates of viral suppression, and NHL risk is high in HIV positive individuals on ART even when their disease is controlled.[14 32 33] This may be because HIV activates the CD40 receptors on B-cells like Epstein-Barr virus (EBV) would in EBV-related cancers such as Burkitt lymphoma.[32] From a national cohort study in South Africa, among AYAs, the incidence of NHL decreased with increasing CD4 cell counts. We expect poor ART coverage and retention in care among AYAs with HIV increases this risk, but researchers still need to determine NHL risk in virally suppressed and non-suppressed patients in our setting. From our interaction analysis, the odds of NHL were higher in adolescents with HIV compared with young adults with HIV. This observation could be as a result of the predominance of lymphoblastic and Burkitt lymphoma, which are more common in younger ages.[34] We also found that the odds of Burkitt lymphoma were higher in HIV positive black AYAs compared with the other ethnicities. Burkitt lymphoma in South Africa is more likely to be found in white children aged 0–14 years than in black children,[22] but this is a different age group to that in our study.

Other studies identified an association between leiomyosarcoma and HIVs.[13] Although not statistically significant, the odds of leiomysorcoma were higher in AYALHIV than in AYAs without HIV. Since leiomyosarcoma is rare, the association between leiomyosarcoma and HIV needs further study. Likewise, after we adjusted for the interaction of HIV with age and calendar period, AYALHIV had an increased risk of connective tissue cancer, but this finding did not reach statistical significance.

We also evaluated HIV unknowns. In South Africa, HIV testing uptake is lower in AYAs than in adults[27] and is mostly opportunistic.[35] Therefore, including this would again stress the importance of patients with cancer and AYAs as a whole to be tested for HIV. Although the percentage of subjects with unknown HIV status decreased over calendar periods, HIV testing for AYA diagnosed with HIV-related cancers remained low. The HIV status of many AYAs with KS, CC and NHL was unknown. An AYA is most likely to be tested if they present to a healthcare

**Table 2** Relationship between HIV and selected cancers among AYAs in the South African public health sector

| Cancer site | Univariable analyses (n=3819) OR (95% CI) | Multivariable complete-case analyses (n=3672) OR (95% CI) | Multivariable imputed analyses (n=8103) OR (95% CI) |
|---|---|---|---|
| AIDS defining cancers | 13.3 (11.2 to 15.8) | 12.0 (9.92 to 14.5) | 11.8 (9.75 to 14.3) |
| Kaposi's sarcoma | 288 (119 to 696) | 218 (89.9 to 530) | 208 (83.9 to 519) |
| NHL | 1.64 (1.34 to 2.00) | 2.12 (1.69 to 2.66) | 2.12 (1.64 to 2.74) |
| Burkitt lymphoma | 1.99 (1.30 to 3.05) | 2.65 (1.64 to 4.28) | 2.78 (1.75 to 4.37) |
| NHL- NOS | 3.31 (1.72 to 6.37) | 4.28 (2.06 to 8.92) | 4.00 (1.78 to 8.99) |
| DLBCL- NOS | 1.52 (1.11 to 2.09) | 2.03 (1.42 to 2.90) | 1.97 (1.34 to 2.91) |
| DILBCL- NOS | 3.61 (1.64 to 7.97) | 4.75 (2.01 to 11.3) | 4.80 (2.04 to 11.3) |
| Mature T-cell- NHL | 0.94 (0.36 to 2.44) | 1.11 (0.38 to 3.23) | 1.05 (0.37 to 2.97) |
| Follicular NHL | 0.79 (0.18 to 3.55) | 0.93 (0.12 to 6.89) | 0.96 (0.09 to 9.80) |
| Cervical cancer | 4.62 (2.75 to 7.75) | 2.18 (1.23 to 3.89) | 2.70 (1.31 to 5.53) |
| Non-AIDS defining cancer | 0.09 (0.08 to 0.11) | 0.11 (0.09 to 0.13) | 0.11 (0.09 to 0.12) |
| Virus-related cancers | 0.56 (0.46 to 0.68) | 0.64 (0.52 to 0.80) | 0.61 (0.48 to 0.79) |
| Anogenital cancers other than cervix | 3.91 (2.00 to 7.65) | 2.73 (1.27 to 5.86) | 2.61 (1.11 to 6.11) |
| Hodgkin's lymphoma | 0.48 (0.38 to 0.60) | 0.60 (0.47 to 0.78) | 0.58 (0.44 to 0.78) |
| Liver and bile duct | 0.27 (0.14 to 0.55) | 0.28 (0.13 to 0.61) | 0.26 (0.11 to 0.64) |
| Virus-unrelated non-AIDS defining cancers | 1.88 (1.57 to 2.25) | 1.69 (1.38 to 2.07) | 1.70 (1.33 to 2.17) |
| Bone | 0.23 (0.16 to 0.32) | 0.29 (0.21 to 0.42) | 0.29 (0.20 to 0.41) |
| Brain-CNS | 0.33 (0.21 to 0.53) | 0.35 (0.20 to 0.60) | 0.37 (0.17 to 0.80) |
| Colorectal | 0.25 (0.14 to 0.44) | 0.15 (0.08 to 0.28) | 0.15 (0.07 to 0.29) |
| Connective tissue | 0.41 (0.30 to 0.57) | 0.44 (0.31 to 0.64) | 0.46 (0.31 to 0.69) |
| Eye | 1.85 (1.05 to 3.27) | 1.11 (0.58 to 2.13) | 1.03 (0.45 to 2.32) |
| Haematology | 0.33 (0.18 to 0.58) | 0.63 (0.34 to 1.18) | 0.67 (0.33 to 1.36) |
| Leukaemia | 0.22 (0.18 to 0.27) | 0.29 (0.23 to 0.37) | 0.30 (0.23 to 0.39) |
| Leiomyosarcoma | 3.18 (0.64 to 15.8) | 2.13 (0.38 to 11.9) | 2.01 (0.31 to 12.9) |
| Myeloma | 0.79 (0.18 to 3.55) | 0.62 (0.09 to 4.03) | 0.63 (0.07 to 5.57) |
| Ovary | 0.51 (0.30 to 0.87) | 0.48 (0.26 to 0.87) | 0.58 (0.31 to 1.08) |
| SCC skin | 1.99 (1.06 to 3.74) | 1.21 (0.60 to 2.44) | 1.07 (0.50 to 2.27) |
| Skin | 0.61 (0.29 to 1.29) | 0.44 (0.19 to 1.02) | 0.37 (0.15 to 0.87) |
| Thyroid | 0.46 (0.19 to 1.12) | 0.29 (0.11 to 0.77) | 0.31 (0.09 to 1.05) |
| Testis | 0.19 (0.08 to 0.42) | 0.28 (0.11 to 0.68) | 0.21 (0.07 to 0.57) |

The multivariable analyses adjusted for age, sex (where applicable), ethnicity and calendar period. Imputation was done under the missing at random assumption for each cancer type. The variables used to impute for missing HIV status were ethnicity, sex and cancer diagnosis year. The imputed analyses are multivariable analysis adjusting for age, sex (where applicable), ethnicity and calendar period.

AYAs, adolescents and young adults; CNS, central nervous system; DILBCL, diffuse immunoblastic large B-cell lymphoma; DLBCL, diffuse large B-cell lymphoma; NHL, non-Hodgkin's lymphoma; NOS, not otherwise specified; SCC, squamous cell carcinoma.

facility with symptoms linked to a sexually transmitted infection or if a female AYA visits a reproductive health clinic.[36]

Because AYALHIV are at higher risk of ADCs and anogenital cancers and many AYAs with HIV-related cancers are not tested for HIV, HIV programmes for AYAs should extend HIV testing coverage, link AYAs to care and make sure to retain them. AYALHIV have a higher risk of cervical and other anogenital cancers because of the high frequency of HPV coinfection, exacerbated by sexual debut and young age. In addition, we also recommend that physicians maintain a high suspicion index

for lymphomas and take care no to misdiagnose them as tuberculosis, thereby delaying care.

Our study is the first nationwide study to compare the distribution of cancers in AYAs by HIV status in South Africa. The record linkage and the additional results determined from the text mining process ensured that we extracted the maximum available HIV results. Our study has several limitations. As in other HIV cohort studies[37 38] that have used CD4 cell counts to create HIV cohorts, we assumed that anyone who had a CD4 cell count test was HIV positive. It is possible, that CD4 cell count tests might be performed for other reasons. We think that this

possibility is low in our study setting, because according to South African management guidelines,[39] since CD4 tests are usually administered after a positive HIV test. The proportion of patients whose HIV status was unknown might not be representative of the entire HIV population in South Africa, because our study included only those who had laboratory HIV tests. Rapid test results are less likely to appear in the NHLS database in the later years compared with the beginning of our study period (only 10% of cancers had a rapid test result). Our study shares the same limitations as the proportionate incidence analysis. Since out study population included only AYAs with cancer just like in proportionate incidence analysis, the ORs may have been overestimated. For the most common cancers, the OR might reflect how frequently a cancer is observed and not the actual strength of association between HIV and the cancer. There is also a potential of a type one error as a result of multiple hypothesis testing on the same data set for the different cancers. However, our results are generally in line with what has been observed in adults in South and AYAs in other settings. Using all other cancers as a comparison group may have also led to underestimating the strength of the association, especially for cancers with overlapping risk factors. In addition, this limits generalisability of our findings to the South African population. However, this does not necessarily mean that the effects of the last two limitations cancel out. Our study was not designed to assess associations between markers of immunosuppression and cancer risk. In our study, HIV negative individuals do not have CD4 cell count measurements and could, therefore, not be in included for such comparisons. Each patient with cancer was assigned only one HIV-related test. Therefore, although we used CD4 cell counts to assign HIV status, we did not assess the sequence of CD4 cell counts and hence cannot establish whether the values were the baseline CD4 cell measurements or the most recent CD4 cell measurements. Lastly, those assigned HIV status using other tests would not have a CD4 cell count, which, would then result in a selection bias. Because of these reasons we did not adjust for markers of immunosuppression such as HIV RNA viral loads and CD4 cell counts in our analyses. We were also unable to assess the odds of cancer by HIV transmission route, for example, vertical transmission against other transmission routes. Finally, the HPV vaccine has been rolled out in 2014 to girls between the ages of 9 and 13 years, therefore, we are not able to evaluate the impact of HPV vaccination on cancer risk with our data that cover the time period 2011–2014.

## CONCLUSION

This is the first nationwide study in South Africa to compare the distribution of cancers in AYAs by HIV status. ADCs and anogenital cancers other than cervix cancer were more common in HIV positive than in HIV negative AYAs. AYAs with cancer are a key population for HIV testing, however, this study suggests that many AYAs with ADCs are not tested for HIV. Targeted HIV testing for AYAs should be followed by the immediate start of ART after a positive HIV diagnosis, accompanied by screening for cervical pre-cancer and vaccination against HPV to decrease cancer burden in AYALHIV in South Africa.

**Author affiliations**
[1]National Cancer Registry, National Health Laboratory Service, Johannesburg, Gauteng, South Africa
[2]Institute of Social and Preventive Medicine, University of Bern, Bern, Switzerland
[3]Swiss Tropical and Public Health Institute, Basel, Switzerland
[4]University of Basel, Basel, Switzerland
[5]Division of Epidemiology and Biostatistics, University of the Witwatersrand School of Public Health, Johannesburg, South Africa
[6]Centre for Infectious Disease Epidemiology and Research (CIDER), University of Cape Town School of Public Health and Family Medicine, Observatory, Western Cape, South Africa
[7]Population Health Sciences, Bristol Medical School, University of Bristol, Bristol, UK
[8]Paediatric Haematology Oncology, Chris Hani Baragwanath Academic Hospital, University of the Witwatersrand, Johannesburg-Braamfontein, Gauteng, South Africa
[9]South African DSI-NRF Centre for Excellence in Epidemiological Modelling and Analysis (SACEMA), Stellenbosch University, Stellenbosch, South Africa

**Acknowledgements** The authors would like to thank all funders, the University of the Witwatersrand, the National Health Laboratory Service (NHLS), the NHLS's Corporate Data Warehouse (special thanks to Sue Candy) and the National Cancer Registry. We would also like to thank Kali Tali for her editorial input.

**Contributors** ME, ES, MM and JB contributed towards the study design. TD contributed towards literature search, data analysis and drafting of first version of manuscript. ES and MM contributed towards data acquisition. AS contributed towards data linkage. VO contributed towards text mining of cancer pathology reports to assign HIV status. MM and TD are the guarantors of the work. All authors contributed towards data interpretation and critical comments on the first and subsequent drafts of the manuscript. All authors read and approved the final manuscript.

**Funding** This work was supported by grants from the US Civilian Research & Development Foundation (CRDF) Global, the National Institutes of Health administrative supplement to Existing NIH Grants and Cooperative Agreements (Parent Admin Supp) (The South African HIV Cancer Match Study; U01AI069924-09, PI ME, co-PI JB) PEPFAR supplement (PI ME), the Swiss National Science Foundation (The South African HIV cancer Match Study, 320030-169967, PI JB). ME was supported by special project funding (grant 17481) from the Swiss National Science Foundation. TD is supported by the European Union's Horizon 2020 research and innovation programme under the Marie Skłodowska-Curie grant agreement No 801076, through the SSPH +Global PhD Fellowship Programme in Public Health Sciences (GlobalP3HS) of the Swiss School of Public Health.

**Competing interests** None declared.

**Patient consent for publication** Not applicable.

**Ethics approval** Permission to use the routinely collected NHLS and NCR data was sought from the relevant authorities. Ethical approval to conduct the study was granted by the University of the Witwatersrand Human Research Ethics Committee [Ethics certificate numbers (SAM: M160944) and (BCAH: M171083)].

**Provenance and peer review** Not commissioned; externally peer reviewed.

**Data availability statement** Data are available on reasonable request. The datasets used and/or analysed during the current study are available from the corresponding author upon reasonable request.

terminology, drug names and drug dosages), and is not responsible for any error and/or omissions arising from translation and adaptation or otherwise.

**ORCID iDs**
Tafadzwa Dhokotera http://orcid.org/0000-0002-4353-0787
Matthias Egger http://orcid.org/0000-0001-7462-5132
Jabulani Ronnie Ncayiyana http://orcid.org/0000-0002-2151-1251
Victor Olago http://orcid.org/0000-0002-0154-0688
Marcel Zwahlen http://orcid.org/0000-0002-6772-6346
Mazvita Muchengeti http://orcid.org/0000-0002-1955-923X

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
