## [Reviewer comments · BMJ Open]

ARTICLE DETAILS

TITLE (PROVISIONAL)	Cancer in HIV-positive and HIV-negative adolescents and young adults in South Africa: a cross-sectional study
AUTHORS	Dhokotera, Tafadzwa; Bohlius, Julia; Egger, Matthias; Spoerri, Adrian; Ncayiyana, Jabulani; Naidu, Gita; Olago, Victor; Zwahlen, Marcel; Singh, Elvira; Muchengeti, Mazvita

VERSION 1 – REVIEW

REVIEWER	Gonçalves, Ana Universidade Federal do Rio Grande do Norte
REVIEW RETURNED	04-Oct-2020

GENERAL COMMENTS	We appreciate the opportunity to collaborate with this prestigious journal reviewing the article: Manuscript: BMJ open-2020-043941entitled “Cancer in HIV-positive and HIV-negative adolescents and young adults in South Africa: a cross-sectional study.” After reading the article and evaluating the paper personally, we feel that the manuscript is a well-written article and needs minor revisions before it can be considered appropriate for potential publication. The comments are included at the bottom of this letter. Abstract/Introduction The acronym AYAs (adolescents and young adults) was not defined previously in the manuscript (abstract and introduction). Title, Objective, and Discussion The authors assumed as a limitation that a CD4 count test indicates being HIV positive but CD4 testing maybe .performed for other reasons. Thus, perhaps the correct thing would be to state that the aim was to determine the spectrum of cancers in AYAs living with HIV and immunosuppressed by unknown causes. However, this limitation was well mentioned in the text, and it is a good quality study. I particularly recommend publishing.
---

REVIEWER	Mukhtar, Fahad Saint Elizabeths Hospital
REVIEW RETURNED	11-Apr-2021

GENERAL COMMENTS	The authors performed a cross-sectional analysis to determine the association between HIV/AIDS in adolescents and young adults living in South Africa using national data from National Cancer Registry and the National Health Laboratory Service. I have several issues that will require major revision. Please see my comments in the attached manuscript. Major issues involve the problem of multiple tests that were performed and the risk of type 1 error as well as the method of handling missing data. I also feel
---

	that the results from the MI and main analysis are significantly different and warrant further analysis and discussion. Other issues are indicated in the manuscript. The reviewer provided a marked copy with additional comments. Please contact the publisher for full details.
--	--

REVIEWER	Salters, Kate Simon Fraser University, Faculty of Health Sciences
REVIEW RETURNED	16-Apr-2021

GENERAL COMMENTS	This is an important paper, addressing a gap in literature assessing both 1) cancer risk among youth with HIV; and 2) cancer risk in a hyper-endemic HIV setting. The authors give a very clear, descriptive analysis demonstrating the burden of cancer among AYA in South Africa. I have a few questions and a few suggestions for the revised paper. My questions stem from what information could further add important clinical information to this descriptive analysis:  -The result that ADC are higher among AYA with HIV (vs without HIV) could be bolstered by additional information. It is clear that looking at CD4 cell count (and viral load) could be a very important way to better understand how HIV (specifically, uncontrolled HIV) is driving cancer risk in AYA in this setting (vs just looking at HIV-status alone). Could the authors include a model that accounts for CD4 cell count over time? This would greatly improve the analysis and strengthen the paper. It is not discussed in the limitations why clinical and therapeutic data were not included in this analysis. -Is there a way to stratify and/or include data on AYA who acquire HIV via vertical transmission vs other routes? This could help account for variances in cancer risk (i.e. highlight potential gaps in treatment leading to cancer risk)? Suggested edits:  -there are several small typos throughout that could be easily fixed -in the introductions, the authors make a good case that ADC are higher among AYA with HIV (vs HIV-negative AYA), but then there needs to be further justification for what this paper adds to the literature. What are the consequences of this increased burden of disease and/or what can be done to mitigate these risks (i.e. estimates of the role of ART). -the authors refer to this study design as a cross-sectional study, but based on their use of registry data, is it not a retrospective cohort study design? -do the authors have any information on staging data? ART status? VL or CD4 over time? -the HIV-status unknown is a confusing part of this analysis, why not exclude unknown status? -ethnicity data: (what does 'coloured' refer to?) -in table 1: it would be helpful to have an 'overall' column and also include the 'n' at the top of each column. In the same table, it would be helpful to have the type of cancer listed in the column (just frequencies of NADC, ADC, viral etc...) -in some cases of cancer, there are small numbers, could you include these numbers (frequencies) overall in table 2? -what was done with AYA in the sample that had a record of cancer prior to HIV diagnosis? -limitations discussed on page 12, should be moved to the end of the discussion section
---

	-the authors note that in their analysis, proportion of KS was higher among girls and young women (vs boys). Why is this? I imagine its because girls/young women have poorer clinical outcomes/barriers to care and it is an indicator of lack of ART access, but could the authors confirm/expand? -authors note that cervical cancers are high and suggest why this may be, but other HPV-related cancers appear to be higher in the sample as well, a discussion of HPV vaccinations/risks could benefit -are there other reasons that HIV-status may be unknown (i.e. just not recorded in this registry but noted in other clinical charts)?
--	---

VERSION 1 – AUTHOR RESPONSE

Reviewer: 1

Dr. Ana Gonçalves, Universidade Federal do Rio Grande do Norte

Comments to the Author:

We appreciate the opportunity to collaborate with this prestigious journal reviewing the article:

Manuscript: BMJ open-2020-043941 entitled “Cancer in HIV-positive and HIV-negative adolescents and young adults in South Africa: a cross-sectional study.”

After reading the article and evaluating the paper personally, we feel that the manuscript is a well-written article and needs minor revisions before it can be considered appropriate for potential publication.

The comments are included at the bottom of this letter.

Abstract/Introduction

The acronym AYAs (adolescents and young adults) was not defined previously in the manuscript (abstract and introduction).

Authors’ response: Many thanks for your observation. We have now defined the acronym in the abstract and introduction of the revised version of the manuscript.

Title, Objective, and Discussion

The authors assumed as a limitation that a CD4 count test indicates being HIV positive but CD4 testing maybe .performed for other reasons. Thus, perhaps the correct thing would be to state that the aim was to determine the spectrum of cancers in AYAs living with HIV and immunosuppressed by unknown causes. However, this limitation was well mentioned in the text, and it is a good quality study. I particularly recommend publishing.

Authors’ response: Many thanks for your comments. We do agree that CD4 cell counts are not only an indication of immunosuppression due to HIV but also other diseases. However, the CD4 counts were not all indicating immunosuppression, there were also measurements within normal reference ranges. In addition, in South Africa, CD4 count testing is mostly indicated for HIV monitoring, after an HIV infection was diagnosed with an HIV test. A study in South Africa validated the use of CD4 cell counts in HIV linkage studies.¹ Therefore, we would still like to maintain that in our study they were all HIV related but still acknowledge the limitation of this assumption in the manuscript.

“Our study has several limitations. As in other HIV cohort studies,^{1,2} that have used CD4 cell counts to create HIV cohorts, we assumed that anyone who had a CD4 cell count test was HIV positive. It is possible, that CD4 cell count tests might be performed for other reasons. We think that this possibility is low in our study setting, because according to South African management guidelines,³CD4 tests are usually administered after a positive HIV test. The proportion of patients whose HIV status was unknown might not be representative of the entire HIV population in South Africa, because our study

included only those who had laboratory HIV tests.”

Reviewer: 2

Dr. Fahad Mukhtar, Saint Elizabeths Hospital

Comments to the Author:

The authors performed a cross-sectional analysis to determine the association between HIV/AIDS in adolescents and young adults living in South Africa using national data from National Cancer Registry and the National Health Laboratory Service. I have several issues that will require major revision. Please see my comments in the attached manuscript. Major issues involve the problem of multiple tests that were performed and the risk of type 1 error as well as the method of handling missing data. I also feel that the results from the MI and main analysis are significantly different and warrant further analysis and discussion. Other issues are indicated in the manuscript.

Authors' response: Many thanks for your comments. If we understood correctly, the multiple tests refer to the statistical analyses. We do agree that there is a potential of type one error resulting from the individual logistic regression for each cancer using the same dataset. However, our results are similar to what has been observed in adults in South Africa as well as AYAs in other settings in previous studies. We have however added a limitations statement in the strengths and limitations paragraph as follows:

‘There is also potential of a type one error as a result of multiple hypothesis testing on the same data set for the different cancers.⁴¹ However, our results are generally in line with what has been observed in adults in South Africa and AYAs in other settings.’

With regards to the multiple imputation, we reviewed the imputations and have now changed how we set up our imputation models. We now impute the missing HIV status for each cancer separately, so having coherent alignment of the model for the substantive analysis and the imputation model.⁴ As a result, the estimated odds ratios from the imputed data are now closer to the complete case analysis. The new estimates are presented in Table 2.

Reviewer: 3

Dr. Kate Salters, Simon Fraser University, British Columbia Centre for Excellence in HIV/AIDS

Comments to the Author:

This is an important paper, addressing a gap in literature assessing both 1) cancer risk among youth with HIV; and 2) cancer risk in a hyper-endemic HIV setting. The authors give a very clear, descriptive analysis demonstrating the burden of cancer among AYA in South Africa. I have a few questions and a few suggestions for the revised paper.

My questions stem from what information could further add important clinical information to this descriptive analysis:

-The result that ADC are higher among AYA with HIV (vs without HIV) could be bolstered by additional information. It is clear that looking at CD4 cell count (and viral load) could be a very important way to better understand how HIV (specifically, uncontrolled HIV) is driving cancer risk in AYA in this setting (vs just looking at HIV-status alone). Could the authors include a model that accounts for CD4 cell count over time? This would greatly improve the analysis and strengthen the paper. It is not discussed in the limitations why clinical and therapeutic data were not included in this analysis.

Authors' response:.. Indeed CD4 cell counts and HIV RNA viral loads would be an essential in understanding cancer in people living with HIV. However, each patient was assigned one HIV test

using different HIV diagnostic and monitoring tests as proxies for HIV positivity. For other individuals the indication of HIV positivity were ELISA, western blot and rapid tests among other methods as well as the viral load counts. In addition, HIV negative patients would not have CD4 cell counts for comparison purposes. The parent study, the South African HIV Cancer Match study is in a better position to evaluate the effect of CD4 cell counts on cancer risk as it is a cohort study.

We include this statement in the limitations section:

“Our study was not designed to assess associations between markers of immunosuppression and cancer risk. In our study, HIV negative individuals do not have CD4 cell count measurements and could therefore not be included for such comparisons. Each cancer patient was assigned only one HIV-related test. Therefore, although we used CD4 cell counts to assign HIV status, we did not assess the sequence of CD4 cell counts and hence cannot establish whether the values were the baseline CD4 cell measurements or the most recent CD4 cell measurements. Lastly, those assigned HIV status using other tests would not have a CD4 cell count, which would then result in a selection bias. Because of these reasons we did not adjust for markers of immunosuppression such as HIV RNA viral loads and CD4 cell counts in our analyses.”

-Is there a way to stratify and/or include data on AYA who acquire HIV via vertical transmission vs other routes? This could help account for variances in cancer risk (i.e. highlight potential gaps in treatment leading to cancer risk)?

Authors' response: Unfortunately, we do not have any further clinical information like vertical transmission of HIV. We have now addressed this in the limitations section

“We were also unable to assess the odds of cancer by HIV transmission route, for example vertical transmission against other routes.”

Suggested edits:

-there are several small typos throughout that could be easily fixed

Authors' response: Thank you for pointing this out, the revised manuscript was carefully micro-edited to avoid typos.

-in the introductions, the authors make a good case that ADC are higher among AYA with HIV (vs HIV-negative AYA), but then there needs to be further justification for what this paper adds to the literature. What are the consequences of this increased burden of disease and/or what can be done to mitigate these risks (i.e. estimates of the role of ART).

Authors' response: We have included the following statement in paragraph 2 of the introduction.

“Estimating the relationship between cancer and HIV is important to estimate their additional health care needs and to provide a baseline for potential mechanisms for prevention of cancer development in AYALHIV.”

-the authors refer to this study design as a cross-sectional study, but based on their use of registry data, is it not a retrospective cohort study design?

Authors' response: Our study is a cross-sectional study that evaluates cancer by HIV status in AYA. Our study does largely not include longitudinal data and we evaluate cancer by HIV status without regard to temporality (included HIV positives before and after HIV diagnosis) therefore it is cross sectional. In our cross-sectional study, most records for HIV and cancer were linked using the episode number. The episode number refers to the tests that are requested for one patient at the same time on the same day. The health care provider usually assigns them the same tracking number. Therefore, this episode is a reflection of a cancer diagnosis and an HIV diagnosis on the

same day. For other patients information on HIV status was extracted from the clinical history section of pathology reports using text mining methods therefore the HIV diagnosis date is unknown.. Lastly, for patients with CD4 cell count measurements as indirect proof for HIV positivity, we lack HIV tests and date of test done before the actual CD4 monitoring tests. We opted to use these methods as they allowed us to have an HIV-negative control group. However, the methods chosen preclude the construction of a cohort. Please note that the parent study, the South African HIV Cancer Match Study, used different methods, allowing to construct a cohort study with repeated HIV monitoring measurements and follow-up time for people living with HIV. That parent study is therefore tailored to estimate incidence rates and risk factors. However, in that cohort study it was no possible to include HIV negative controls, as we do not have laboratory records for HIV negative persons that would allow us to measure time under observation and create an actual cohort.

We have rephrased it in the methods section for better understanding as follows:

“For the deterministic record linkage we used episode numbers as linkage variable. Episode number refers to tests that were requested for a patient at the same time by the health practitioner and assigned the same unique identifier. About 65% of the all linkages were matched using the episode number.”

-do the authors have any information on staging data? ART status? VL or CD4 over time?

Authors' response: Unfortunately, data on staging and, ART status is not available for this analysis. Our study was designed to give a cross-sectional comparison of cancer in HIV positive and negative AYAs with each cancer being assigned 1 HIV record. CD4 cell counts were only used to assign HIV status and having a CD4 cell count was not systematic for everyone. In addition, for each patient we would have only one CD4 cell count and there would be no way to ascertain whether this is the baseline CD4 cell count or the most recent one. On the other hand, the main study, the South African HIV cancer match (SAM) study involves an HIV cohort created from HIV diagnostic and monitoring test. There are no HIV negative individuals in that study. The SAM study is better positioned to determine the relationship between cancer and CD4 cell counts but without the HIV negative comparisons.

-the HIV-status unknown is a confusing part of this analysis, why not exclude unknown status?

Authors' response: Many thanks for your comment. It is possible we might have missed other individuals who know their HIV status such as those who had point of care tests in the later years that were not available in the NHLS dataset. We believe HIV unknown holds important information as it stresses the importance of being tested for HIV. For example, with Kaposi sarcoma, we expect to see very little missing HIV results but there is still a substantial proportion of HIV related cancers with an unknown HIV status. We discuss this in detail in the paper as follows;

“We also evaluated HIV unknowns. In South Africa, HIV testing uptake is lower in AYAs than in adults⁵ and is mostly opportunistic.⁶ Therefore, including HIV unknowns would again stress the importance of cancer patients and AYAs as a whole to be tested for HIV. Although the proportion of subjects with unknown HIV status decreased over calendar periods, HIV testing for AYA diagnosed with HIV-related cancers remained low. The HIV status of many AYAs with KS, CC and NHL was unknown. An AYA is most likely to be tested if they present to a health care facility with symptoms linked to a sexually transmitted infection or if a female AYA visits a reproductive health clinic.⁷”

-ethnicity data: (what does 'coloured' refer to?)

Authors' response: The term coloured refers to the mixed race population. This is a commonly used term used in the country including the National Cancer Registry and the formal terminology of our statistical bureau Statistics South Africa. We have also added “mixed race” in brackets.

-in table 1: it would be helpful to have an 'overall' column and also include the 'n' at the top of each column. In the same table, it would be helpful to have the type of cancer listed in the column (just frequencies of NADC, ADC, viral etc...)

Authors' response: Thank you for your comment. We have included an overall column.

-in some cases of cancer, there are small numbers, could you include these numbers (frequencies) overall in table 2?

Authors' response: Many thanks for your comments. We have added the individual cancer frequencies to Table 2

-what was done with AYA in the sample that had a record of cancer prior to HIV diagnosis?

Authors' comments: Since this is a cross sectional study, we included individuals with an HIV diagnosis before or after the cancer diagnosis. Please note about 65% of the cancer and HIV tests were requested on the same day.

-limitations discussed on page 12, should be moved to the end of the discussion section

Authors' comments: Limitations have been moved to the end of the discussion section as suggested.

-the authors note that in their analysis, proportion of KS was higher among girls and young women (vs boys). Why is this? I imagine its because girls/young women have poorer clinical outcomes/barriers to care and it is an indicator of lack of ART access, but could the authors confirm/expand?

Authors' response: Many thanks for your comment. In our study, the proportion of males with known HIV was lower than those of females. Since KS is a known HIV related cancer, the high proportion of KS that we observed in females is likely a reflection of the HIV positivity we had in our population. When it comes to the measures of association which we do not report here, most studies do mention that the risk/ odds are higher for men as their retention in care is poorer.

-authors note that cervical cancers are high and suggest why this may be, but other HPV-related cancers appear to be higher in the sample as well, a discussion of HPV vaccinations/risks could benefit

Authors' response: According to literature the prevalence of HPV including high risk variants is high in young adults with HIV in South Africa⁸ and this links to the increased odds of HPV related cancers. HPV vaccinations were introduced through a school based programme for girls age 9-13 in 2014. Since our data is from 2004-2014, we cannot evaluate the impact of the vaccination within our study period. However, we do make reference to the potential high prevalence of HPV in AYAs and the importance of vaccination to prevent early presentation with HPV related cancers particularly those caused by the subtypes covered by available vaccines.⁹

-are there other reasons that HIV-status may be unknown (i.e. just not recorded in this registry but noted in other clinical charts)?

Authors' response: Thank you for your comment. We acknowledge that there could be many reasons why the HIV tests would be missing. Our consideration was that the individuals with missing HIV tests were not tested at all within our study period. Another possibility is that they might have been tested in the later years using point of care tests that were not available in the NHLS database. In addition to the routinely collected HIV data from the NHLS, we also text mined the cancer pathology reports. This

means we looked at the clinical history section for any information on HIV status. If it was recorded, we also added it to our dataset and this improved the completeness of our study.

VERSION 2 – REVIEW

REVIEWER	Mukhtar, Fahad Saint Elizabeths Hospital
REVIEW RETURNED	30-Jul-2021
GENERAL COMMENTS	The authors' have made all the necessary major changes to this manuscript. I will only recommend reviewing the manuscript for syntax and grammar; perhaps in the course of revision some words may have been omitted. In addition, the limitations paragraph is quite wordy; consider revising and rewording the paragraph to be more concise.